# Persistent Bacterial Infections, Antibiotic Treatment Failure, and Microbial Adaptive Evolution

**DOI:** 10.3390/antibiotics11030419

**Published:** 2022-03-21

**Authors:** Ruggero La Rosa, Helle Krogh Johansen, Søren Molin

**Affiliations:** 1The Novo Nordisk Foundation Center for Biosustainability, Technical University of Denmark, 2800 Kgs. Lyngby, Denmark; rugros@biosustain.dtu.dk (R.L.R.); hkj@biosustain.dtu.dk (H.K.J.); 2Department of Clinical Microbiology 9301, Rigshospitalet, 2100 Copenhagen, Denmark; 3Department of Clinical Medicine, Faculty of Health and Medical Sciences, University of Copenhagen, 2200 Copenhagen, Denmark

**Keywords:** infection persistence, antibiotic resilience, adaptative evolution

## Abstract

Antibiotic resistance is expected by the WHO to be the biggest threat to human health before 2050. In this overview, we argue that this prediction may in fact be too optimistic because it is often overlooked that many bacterial infections frequently ‘go under the radar’ because they are difficult to diagnose and characterize. Due to our lifestyle, persistent infections caused by opportunistic bacteria—well-known or emerging—show increasing success of infecting patients with reduced defense capacity, and often antibiotics fail to be sufficiently effective, even if the bacteria are susceptible, leaving small bacterial populations unaffected by treatment in the patient. The mechanisms behind infection persistence are multiple, and therefore very difficult to diagnose in the laboratory and to treat. In contrast to antibiotic resistance associated with acute infections caused by traditional bacterial pathogens, genetic markers associated with many persistent infections are imprecise and mostly without diagnostic value. In the absence of effective eradication strategies, there is a significant risk that persistent infections may eventually become highly resistant to antibiotic treatment due to the accumulation of genomic mutations, which will transform colonization into persistence.

## 1. Introduction

Bacterial infections have become increasingly difficult to treat with antibiotics, and it has been predicted that infectious diseases will become a bigger challenge to human health in a few decades than, for example, cancer [1]. Antibiotic resistance is considered the major cause of this threat, and it is frequently argued that the solution to this problem is to discover new antibiotics to replace those that are no longer active against resistant bacteria [2]. The major reason for the rise of antibiotic resistance in the world is an inappropriate and severe overuse of antibiotics in human therapy, as well as agriculture for domestic animals [3,4,5,6,7].

However, there are other important reasons for this situation, such as the rapid spread of new emerging pathogens due to intensive global travel and widespread unhealthy lifestyles. An increasing share of all human infections are far from simple and are difficult to cure for reasons other than antibiotic resistance [3]. Moreover, many bloodstream infections derive from persistent bacterial infections, which affect a broad range of people with underlying conditions such as diabetes, obesity, smoking, immune-compromised conditions, implants, etc. [4].

Non-curable persistent infections are often multi-factorial and dynamically adaptive, and have the potential to develop into life-long chronic infections associated with increased morbidity and mortality [5] because they do not follow the traditional medical ‘rules’ for infection, but rather, display the following traits:Difficulty in identification of the infecting pathogen(s).Irrelevance of Koch’s principles.Unreliable antibiotic resistance determinations.Epistatic mechanisms underlying failures to treat with antibiotics.

The strong focus on antibiotic resistance as a major health threat is mainly associated with acute infections by pathogenic bacteria, which have become resistant to one or more antibiotics after passage in previous environments (hospitals, sewage, areas close to drug-producing industrial plants), or through genetic exchange with resistant bacteria in agricultural productions, where antibiotics are used extensively [1,6,7]. Persistent bacterial infections, which are often caused by susceptible opportunistic pathogens from the environment, are difficult to treat due to a combination of an intrinsic high tolerance to many different antibiotics and the special lifestyles of these bacteria in the infected tissues (Table 1). Sustained bacterial survival in the presence of antibiotics inevitably results in resistance developing in the infecting bacterial population, and often the underlying genetic resistance mechanisms are different from those usually observed in the clinic in connection with acute infections.

In the clinic, the number of patients with persistent infections is growing, leaving clinicians with a number of pertinent questions:


*Why do some infections persist, can markers for progressive infection processes be identified, can treatment be improved, and can we develop sustainable treatment strategies with greatly reduced risks of resistance development?*


## 2. Why Do Antibiotics Fail to Eradicate Persistent Bacterial Infections?

Although surveillance of antibiotic resistance is important, it is important to stress that antibiotic susceptibility may vary over a large range of MIC values, depending on the growth conditions, and the standardized laboratory setting for the measurement of antibiotic resistance bears very little resemblance to the local conditions in infected patients [8]. Many external factors may have an impact on antibiotic susceptibility, such as: environmental composition of nutrients and their availability, bacterial growth rate, stress factors, interactions with the host and with other microbes at the infected site, bacterial lifestyle (sessile or planktonic), etc. (Figure 1) [9]. These complicating factors explain why persistent infections are often thought to require aggressive treatment with combinations of antibiotics in the clinic, and why treatment of apparently antibiotic sensitive bacterial populations fails to eradicate the bacteria [10]. Continuous unsuccessful antibiotic treatment will eventually result in genetically determined resistance—not only of the targeted infecting bacteria, but also of commensal bacterial populations residing in other organs and tissues of the patient [11].

Antibiotic tolerance (an epigenetically determined decrease in antibiotic susceptibility) is conditional, depending on many internal and external factors, and usually, it is not specifically associated with the presence of a single gene or the occurrence of a single specific mutation [12]. Tolerance further depends upon the physiological state of the bacteria as determined by environmental conditions, including the levels of nutrients, stress factors, and microbial community members [13]. Infecting bacteria use unconventional mechanisms to overcome antibiotic treatment, such as low-level AMR, which is almost completely overlooked in the diagnostic laboratory, and also, under-researched [14]. Importantly, low-level AMR serves as a steppingstone for developing high-level AMR, and therefore, early diagnosis must be a high priority in order to minimize selection for high-level AMR and to direct therapeutic strategies [15]. Low-level AMR development in persistent infections is a complex phenomenon influenced by several factors that change during the infection. By definition, a strain is low-level resistant when its minimum inhibitory concentration (MIC) is slightly higher (~2-fold increase) than that of the reference strain [14]. However, the effect is not marginal, and indeed, it provides sufficient AMR and low fitness cost to overcome treatment [16]. In contrast to high-level AMR, which can be achieved by targeted mutations, low-level AMR is generally the result of mutations in housekeeping genes, changes in the transcriptional profile or rewiring of the metabolism, which have important effects on the physiology of the cell [17,18], but are challenging to decipher and characterize mechanistically (Figure 1).

Confounding factors, such as (1) low sensitivity and non-specificity of conventional antimicrobial sensitivity testing (AST), (2) synergistic or antagonistic effects on AMR caused by the presence of a polymicrobial infection, and (3) absence of reliable biomarkers for bacterial adaptation, challenge early diagnosis and interventions against low-level resistant bacteria. It has been proposed that metabolites can enhance antibiotic sensitivity by triggering specific metabolic pathways, leading to reactivation of persistent cells. However, a systematic and generic knowledge concerning such connections is largely lacking. Moreover, the role of metabolic adaptation in reducing reactive oxygen species (ROS) dependent cellular stress, caused by the use of many antibiotics, is unclear [19].


*In conclusion, cautious treatment strategies for persistent infections based on the intelligent choice of antibiotics, combined with a clinical determination for the susceptibility of the target microbe, is therefore, at best, a partial solution to the problem, and frequently, infected patients show no clearance at all. There is, therefore, a strong need for more precise and relevant diagnostics and improved treatment designs.*


## 3. Which Genetic and Phenotypic Changes Impact the Persistence of Infecting Bacteria?

Persistent bacterial infections caused by environmental opportunistic pathogens are usually associated with extensive adaptive processes, which shape the bacterial population towards increased fitness and niche specialist phenotypes [20]. Investigations of bacterial isolates from year-long persistent infections (>20 years) have documented how hundreds of mutations have accumulated in the respective genomes, and the repeated findings of numerous mutations in certain genes (patho-adaptive genes) suggests that these mutations are most likely adaptive (Figure 1) [21]. Such findings indicate that it may be possible to identify specific genetic markers predicting the continued persistence of the infection. Thorough analyses of large numbers of clinical isolates from long-term infections have shown, however, that neither single mutations nor simple combinations of a few mutations seem to be useful as reliable predictors of persistence [22]. Nevertheless, patterns of patho-adaptive mutations in the early stages of infection point towards the following classes of mutated genes as important for successful persistence: global regulatory genes, stress response genes (including few antibiotic resistance genes), bacterial adherence associated genes, and metabolic genes [21]. Notably, this diversity of adaptive mutations documents the associated diversity of selection forces in the infected environment, which is important to take into consideration in both diagnostic and therapeutic contexts. When the corresponding phenotypes are investigated, a picture emerges suggesting that the accumulated patho-adaptive mutations secure bacterial colonization in the host through genetic changes affecting energy- and biomass-associated metabolism, as well as tolerance of host-associated stresses (including antibiotics and the immune system) [23,24]. An apparent common phenotypic change among bacterial isolates obtained from persistent infections is a reduced growth rate, which is often associated with an increased tolerance for antibiotics (Figure 1) [25].

Bacterial survival in the presence of antibiotics may be inferred from other types of conditions and causes, which can induce increased antibiotic tolerance in the entire bacterial population, or in specific subpopulations. It has been known for decades that stressed bacterial populations are often more tolerant to many antibiotics than populations residing in more harmonious environments [26]. In some cases, such tolerance can convert the entire population from a fully susceptible state to a highly resilient state, in which even very high local concentrations of antibiotics may have no killing effect at all. The biofilm state of bacteria represents a similar condition, which despite the absence of stress factors induces a high tolerance phenotype, although no specific genetic changes are involved in this type of resilience to treatment [27]. Considering that both host-induced stress and a biofilm state of growth associated with many different persistent bacterial infections seems to be the rule, it is obvious that no genomic analysis is relevant for a diagnosis of this type of antibiotic resilience.

A special—and biologically interesting—case of antibiotic tolerance is connected to the persister phenotype. Persister cells occur in susceptible bacterial populations as one or more subpopulations, which when treated with antibiotics, survive the treatment and then resuscitate when the antibiotic is no longer present in the environment. Importantly, the surviving persister bacteria are still susceptible to the antibiotic when re-treated [28]. Although the mechanism behind the persister phenotype is not fully understood, it seems to be associated with a dormant state of the respective subpopulations. However, it is important to stress that there may be several explanations for the persister phenotype [29]. It is also important to note that the relative size of the persister population may vary a lot, depending upon environmental conditions, and as the size of the persister subpopulation increases in an infected host, it will be more difficult eradicate with antibiotics. This problem is amplified by genetic mutations, which result in increased subpopulation sizes for the persisters during the course of an antibiotic treatment. Such Hip (high-persister) mutations have been described for many bacterial species, and in a few cases, the clinical implications of these mutations have been investigated [30]. For example, it was recently observed that Hip variants of *Pseudomonas aeruginosa* were frequently identified in lung-infecting bacterial populations from cystic fibrosis patients, and their prevalence suggested that such variants have a fitness advantage during continued infection. However, no specific genetic alterations could be associated with the Hip phenotype [31].

Similarly, it was recently observed that bacterial populations growing with antibiotics in the environment display heteroresistance—a phenomenon where subpopulations of the bacteria react to the antibiotics by developing transient increased resistance. One important mechanism behind heteroresistance is the duplication of genes conferring partial antibiotic resistance. In accordance with the clinical relevance described above for persister cells and Hip variants, there are indications that heteroresistance may also result in treatment failures for bacterial infections in the clinic [32].

## 4. Can We Predict Infection Persistence and Resilience to Antibiotics from Genome Sequences?

From the earliest days of molecular microbiology, we have become accustomed to associating changes in bacterial phenotypes with highly specific mutations in single genes. Changes in antibiotic susceptibility have been assumed to be the consequence of genetic changes in one, or a few, specific genes [33]. More recently, it has become clear that horizontal mobility of specific resistance determinants involved in modifying or destroying antibiotics is a highly important additional factor for developing antibiotic resistance due to their rapid epidemic spread in bacterial populations [34]. This realization has already been implemented for the design of diagnostic tools for the discovery of both ‘traditional’ target mutations and horizontally transferred antibiotic resistance determinants, which can detect the presence of these modifications in infecting bacterial isolates with high precision [35]. However, in connection with mutations in the bacterial genome causing decreased antibiotic susceptibility, genome sequence-based detection is, however, more complicated due to the genomic diversity of bacterial genes. Due to genetic drift, random mutations accumulate in all genomes—including genes associated with antibiotic targets and resistance—and it is therefore much more difficult to predict a susceptibility phenotype from genomic sequence information. In some cases, the antibiotic target is encoded by highly conserved DNA sequences and predictions from the genome sequences may be more precise, but in other cases, this is not possible, though the recent application of artificial intelligence approaches has resulted in a promising improvement of the predictions for antibiotic resistance based on genomic screening [36].

It is becoming increasingly clear that antibiotic resistance arising in populations of infecting bacteria may derive from combinations of mutations (‘epistatic mutations’), which occur with high frequencies in hypermutator populations during extended infection periods. Hypermutators are either genetic variants with significantly increased mutation rates, or transient phenotypic variants induced by high levels of stressors such as the oxygen radicals produced by an infected host. Both types of variants are frequently detected in connection with persistent infections, and in most cases, the specific mutations leading to increased antibiotic resistance will be very difficult to identify with genome sequencing due to the increased number of mutations in hypermutator genomes [37].


*In conclusion, it is important to be cautious about genome sequence information as the only, or even major, source of predictive diagnostic information in cases of persistent infection. Although findings of a small number of patho-adaptive mutations provided hope for identifying genetic markers for subsequent chronicity, it seems that the changes in the functionality of such mutations are so complex that, in most cases, no precise consequences can be drawn from the genomics. Instead, complex phenotypic changes such as metabolic re-direction and growth rate reduction may carry a predictive power that is useful for diagnosing the persistence of infections in the clinic.*


## 5. Perspective: Sustainable Treatment of Persistent Bacterial Infections

Sustainable treatment of persistent infections is defined as the targeted antimicrobial removal of pathogenic bacteria causing long-term infection in humans based upon the following criteria:(1)*Improved diagnostics for persistence and identification of dominant pathogens;*(2)*Identification of markers for their potential to develop into a chronic infection state;*(3)*Identification of the precise antimicrobial targets for dominant pathogens;*(4)*Design of antimicrobial agents that are bio-degradable or inert in the environment.*

In summary, as more and more people suffer from persistent bacterial infections due to an increased lifespan and an improved lifestyle, which reduces the efficacy of the immune system, the required extensive and life-long treatments with antibiotics for these infections needs to be reconsidered in the context of the unsustainable global use of antibiotics. It is argued here that the frequent failure of antibiotic treatment for long-term bacterial infections is rooted in (1) poor knowledge about the progression of infection and the associated adaptive processes of the colonizing bacteria, (2) an array of bacterial protection strategies in addition to resistance development, and (3) erroneous antibiotic susceptibility diagnoses, reflecting the difference between a clinical test and the patient’s environmental conditions. In addition, antibiotic treatments still have the unwanted potential for selecting resistance among the surrounding bacterial populations in the patient.

There is an urgent need for translational research in relation to persistent bacterial infections to uncover the biological factors and mechanisms that are responsible for the medical failures concerning directed diagnostics and therapy. While conventional antimicrobial treatment fails to limit persistent infections, new approaches such as the use of metabolites as enhancers for antimicrobials may be the key for re-sensitizing resistant bacteria and, finally, to ease the development of AMR [38]. Moreover, advanced antimicrobial sensitivity testing with higher sensitivity that takes into account the complexity of the host environment may allow the early detection of low-level AMR. Unfortunately, despite their great potential, these elements are currently completely neglected. Due to the complexity of the system, therefore, new approaches are required to uncover how changes in the lifestyle of an infecting bacterium finally leads to persistence in the patient without the development of high-level AMR.

## Figures and Tables

**Figure 1 antibiotics-11-00419-f001:**
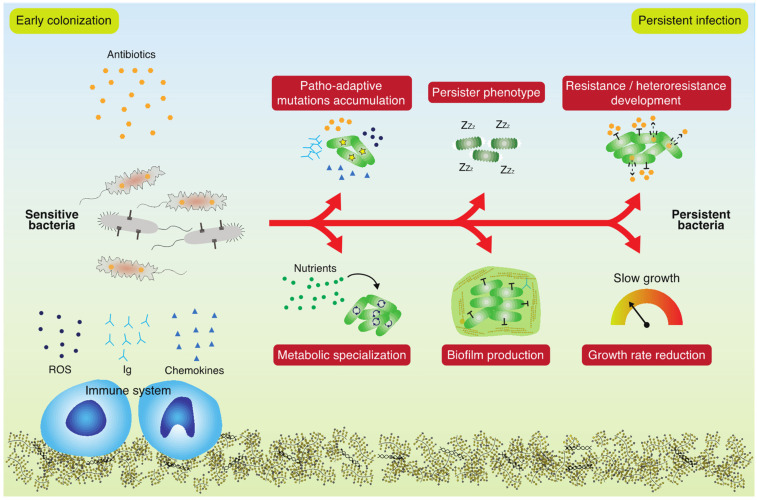
Bacterial adaptation during a persistent infection. Sensitive bacteria are subjected to stresses such as antibiotics and the immune system and use unconventional mechanisms to persist in the host. Patho-adaptive mutations accumulation, persister phenotype, resistance and heteroresistance development, metabolic specialization, biofilm associated lifestyle and growth rate reduction are some of the main mechanisms of adaptation leading to an untreatable persistent infection.

**Table 1 antibiotics-11-00419-t001:** Examples of diseases associated with persistent bacterial infections.

Cystic Fibrosis	*Pseudomonas aeruginosa*, *Achromobacter* spp., *Burkholderia* spp., *Staphylococcus aureus*
Primary Ciliary Dyskinesia	*Haemophilus influenzae*, *S. aureus*, *Moraxella catarrhalis*, *P. aeruginosa*
Chronic Obstructive Pulmonary Disease	*P. aeruginosa*
Orthopedic surgery implants	*S. aureus*, *Cutibacterium* spp., Coagulase negative Staphylococci, *Corynebacterium* spp.
Urinary tract infections	*Escherichia coli*
Implants (vessels, etc.)	*S. aureus*, *Corynebacterium* spp., Coagulase negative Staphylococci
Chronic wounds	*S. aureus*, *P. aeruginosa*, anaerobic bacteria
Stomach ulcers	*Helicobacter pylori*

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
