# Peer review of "Persistent Bacterial Infections, Antibiotic Treatment Failure, and Microbial Adaptive Evolution"

_antibiotics, 2022, doi:10.3390/antibiotics11030419_

Round 1
Reviewer 1 Report
The publication entitled “Persistent Bacterial Infections, Antibiotic Treatment Failure and Microbial Adaptive Evolution” is interesting but some aspects could be taken to the attention of the authors prior to publication:
- The authors suggest lifelong antibiotic therapy, but I have doubts about this approach. It is now believed that frequent antibiotic treatment may be associated with cancer. Bacterial infections result from dysbiosis. Dysbiosis has many sources. One of them is, for example, the low level of vitamin D, stress, disturbed sleep, lack of exercise and improper diet. Treatment with an antibiotic should be combined with a probiotic, which will rebuild the host's commensal microbiota. Have the authors considered the impact of probiotic therapy on persistent bacterial infections and treatment?
- In one of the paragraphs, the authors mention metabolites of pathogenic bacteria. Currently, there are known metabolites of commensal bacteria, such as butyric acid, which may also be helpful in antibiotic therapy. In my opinion, antibiotics, while saving lives, cannot be considered the sole restoring agent to health. Maybe a good way would be to combine several methods, including natural ones?
Author Response
The publication entitled “Persistent Bacterial Infections, Antibiotic Treatment Failure and Microbial Adaptive Evolution” is interesting but some aspects could be taken to the attention of the authors prior to publication:
- The authors suggest lifelong antibiotic therapy, but I have doubts about this approach. It is now believed that frequent antibiotic treatment may be associated with cancer. Bacterial infections result from dysbiosis. Dysbiosis has many sources. One of them is, for example, the low level of vitamin D, stress, disturbed sleep, lack of exercise and improper diet. Treatment with an antibiotic should be combined with a probiotic, which will rebuild the host's commensal microbiota. Have the authors considered the impact of probiotic therapy on persistent bacterial infections and treatment?
I think the reviewer may have misunderstood our message – we do not suggest life-long antibiotic treatment anywhere – we describe the consequences.
- In one of the paragraphs, the authors mention metabolites of pathogenic bacteria. Currently, there are known metabolites of commensal bacteria, such as butyric acid, which may also be helpful in antibiotic therapy. In my opinion, antibiotics, while saving lives, cannot be considered the sole restoring agent to health. Maybe a good way would be to combine several methods, including natural ones?
We actually suggest in the final section that addition of certain metabolites may act as enhancers of antibiotics.
Reviewer 2 Report
Dear Authors!
Thanks for the good solid review on persistent bacterial infections. Written in a correct and appropriate linguistic style it has a straightforward message. The structure of the manuscript is logical, attached figure visualises the content and is well described. The minor comment that could improve the clinical relevance of the paper:
There could be the section/table with examples of persistent bacterial infections (symptomatic or asymptomatic) and well-known or emerging opportunistic bacteria that have a role in it.
Author Response
Dear Authors!
Thanks for the good solid review on persistent bacterial infections. Written in a correct and appropriate linguistic style it has a straightforward message. The structure of the manuscript is logical, attached figure visualises the content and is well described. The minor comment that could improve the clinical relevance of the paper:
There could be the section/table with examples of persistent bacterial infections (symptomatic or asymptomatic) and well-known or emerging opportunistic bacteria that have a role in it.
A Table has now been inserted as suggested by the reviewer.
Reviewer 3 Report
Thank you for giving me the opportunity to review this interesting opinion article by La Rosa et al., an approach to the issue of persistent bacterial infections, antibiotic treatment failure, and microbial adaptive evolution.
There are some comments that I would like to make, with the intention of improving the manuscript:
Perspectives section:
1.- The authors should mention in this section, and include a reference, to the important fact of inappropriate use and overuse of antibiotics as one of the main drivers for the development of microbial resistance.
2.- In addittion, the authors state that "antibiotic treatment of human acute infections driven by susceptible pathogenic bacteria, however, rarely results in direct in-patient resistance development." (Lines 50-51). This statement should be reconsidered and reworded, as there is evidence on the contrary regarding some groups of antibiotics such quinolones and beta-lactams (i.e., Bosso et al., 2006, doi:10.1128/AAC.01359-05; Costelloe et al., 2010, doi:10.1136/bmj.c2096; Cuevas et al., 2014, doi: 10.1093/jac/dkq471). Otherwise, please include references to support that affirmation.
Why do antibiotics fail to eradicate persistent bacterial infections? section:
3.- Lines 90-91: Please add a reference for the definition of low-level resistant strain.
Perspective: Sustainable treatment of persistent bacterial infections section.
4.- Lines 234-235: the state: "uninformed choices of antibiotics for treatment with no or only little effect on the target bacteria due to erroneous antibiotic susceptibility diagnostics in the clinic", could mislead the reader. Do the authors mean that the errors dwell in the Microbiological reports due to not sufficiently well-performed ASTs, or in the interpretation of the ASTs by the physicians? Please, clarify.
Author Response
Thank you for giving me the opportunity to review this interesting opinion article by La Rosa et al., an approach to the issue of persistent bacterial infections, antibiotic treatment failure, and microbial adaptive evolution.
There are some comments that I would like to make, with the intention of improving the manuscript:
Perspectives section:
1.- The authors should mention in this section, and include a reference, to the important fact of inappropriate use and overuse of antibiotics as one of the main drivers for the development of microbial resistance.
I have added a short statement with relevant citations in the beginning of the paper
2.- In addittion, the authors state that "antibiotic treatment of human acute infections driven by susceptible pathogenic bacteria, however, rarely results in direct in-patient resistance development." (Lines 50-51). This statement should be reconsidered and reworded, as there is evidence on the contrary regarding some groups of antibiotics such quinolones and beta-lactams (i.e., Bosso et al., 2006, doi:10.1128/AAC.01359-05; Costelloe et al., 2010, doi:10.1136/bmj.c2096; Cuevas et al., 2014, doi: 10.1093/jac/dkq471). Otherwise, please include references to support that affirmation.
I have removed the sentence and reshaped the argument
Why do antibiotics fail to eradicate persistent bacterial infections? section:
3.- Lines 90-91: Please add a reference for the definition of low-level resistant strain.
The ref has now been inserted
Perspective: Sustainable treatment of persistent bacterial infections section.
4.- Lines 234-235: the state: "uninformed choices of antibiotics for treatment with no or only little effect on the target bacteria due to erroneous antibiotic susceptibility diagnostics in the clinic", could mislead the reader. Do the authors mean that the errors dwell in the Microbiological reports due to not sufficiently well-performed ASTs, or in the interpretation of the ASTs by the physicians? Please, clarify.
I have changed the sentence – hopefully it is more clear now